# Leveraging Complementary Resources through Relational Capital to Improve Alliance Performance under an Uncertain Environment: A Moderated Mediation Analysis

Xian Liu [1,*] , Wenyu Wang [1] and Yiyi Su [2,*]

1   School of Tourism, Hunan Normal University; Changsha 410081, China
2   School of Economics and Management, Tongji University, Shanghai 200092, China
*   Correspondence: jane_liu323@hunnu.edu.cn or jane_liu323@hotmail.com (X.L.); suyiyi@tongji.edu.cn (Y.S.); Tel.: +86-0731-88872154 (X.L.); +86-021-65983573 (Y.S.)

**Abstract:** Integrating the resource-based theory, the relational view, and the contingency theory, this research advances the strategic alliance literature by providing a theoretical framework that explains alliance outcomes from both the inter-organizational and the external environmental perspectives. Specifically, we analyzed the effect of complementary resources on alliance performance through the mechanism of relational capital in an uncertain environment. We also explored the moderating roles of environmental dynamism and environmental hostility in the indirect relationship between resource complementarity and alliance performance with a moderated mediation model. Based on the empirical evidence from a survey of 210 alliance firms, we found that complementary resources that a firm can access from its strategic alliance motivate it to invest relational capital in the partnership, which in turn improves alliance performance. However, the positive link between resource complementarity and relational capital is attenuated under a highly dynamic environment. More importantly, results of the moderated mediation analysis suggest that the mediating effect of relational capital between resource complementarity and alliance performance is stronger when the environment is less dynamic, but this effect is not moderated by environmental hostility. These findings imply that complementary resources are critical antecedents of alliance performance, yet firms could not leverage the value of complementary resources to achieve alliance success without sound relational mechanisms or the ability to adapt to the uncertain environment.

**Keywords:** strategic alliance; environmental uncertainty; complementary resource; relational capital; alliance performance





## 1. Introduction

In recent years, strategic alliances have been widely adopted as an important strategic vehicle for companies to access valuable resources, reduce transaction costs, exploit new opportunities and achieve organizational sustainability. Strategic alliances refer to voluntary arrangements between firms involving the exchange, sharing, or co-development of products, technologies, or services [1]. Despite an increasing number of strategic alliances, research has shown that the failure rate of strategic alliances often exceeds 50 percent [2,3]. A variety of theories provide different explanations for the formation and evolution of strategic alliances. Among these theories, the resource-based view (RBV) asserts that alliance outcomes are based on the effective integration of the alliance firms' unique and non-overlapping resources [4–6]. According to RBV, firms are motivated to search for partners with valuable and complementary resources that create synergy and collective strengths to maintain a competitive advantage [4,7]. In contrast, the relational view (RV) argues that strategic alliances are often threatened by high relational risks that may hinder the achievement of cooperative objectives [8]. This stream of research highlights the influence of relational capital, such as communication, trust, and commitment between

partners, on alliance performance [9–11]. Although both RBV and RV have been influential in predicting alliance performance, they mainly focus on factors in a particular phase of strategic alliances. The RBV attaches great importance to resource characteristics in the pre-alliance and partner selection phase, while RV emphasizes relational governance during the post-alliance formation and alliance management phase. We argue that strategic alliances should be treated as a dynamic process, and thus alliance outcomes depend on how well firms manage factors in each stage of the alliance lifecycle. Additionally, RBV provides insights into the structural aspects of alliance partners whereas RV deals with sociopsychological issues in the cooperation. Some scholars call for complementing the structural approach with the sociopsychological approach to have a comprehensive understanding of alliance formation and outcome [12–14]. As a response to this call, we identified the relationship between RBV and RV, and incorporated these two views to investigate the drivers of alliance performance.

The above theories pay attention to the inter-organizational elements of strategic alliances, the contingency theory instead suggests that the external environment has a great impact on organizational structure and functioning [15–17]. Contingency theories claim that the efficacy of a firm's strategy is linked to various environmental or contextual factors, and the performance is a function of the congruence between the organization and the external environment [18,19]. Drawing on the contingency theory, it is reasonable to analyze alliance performance from both the inter-organizational and the external environmental perspectives. In particular, environmental uncertainty arising from rapid technological changes, turbulent financial and capital markets, increasingly fierce competition, or unpredictable public health crises (e.g., the COVID-19 pandemic) poses huge threats to alliance firms. This research extends the contingency theory and the environmental uncertainty literature by examining the effects of two types of environmental uncertainty, namely environmental dynamism and environmental hostility, on inter-firm relationships and alliance performance.

Given the complexity of alliance structures and factors, it is difficult to fully understand strategic alliances with a single theory [11]. Our research addresses these gaps in knowledge by integrating the resource-based view, the relational view, and the contingency theory to provide an integrated framework for predicting alliance performance. Specifically, we examine how complementary resources affect alliance performance through the mediating effect of relational capital in an uncertain environment. We also examine the boundaries of the mediating role of relational capital between resource complementarity and alliance performance by exploring the moderating effects of environmental dynamism and environmental hostility with a moderated mediation model. It is important to explore these effects because leveraging and managing complementary resources and relational capital to aid the success of an alliance in an environment with ever-increasing uncertainty is receiving considerable attention from both management scholars and business practitioners.

In the following sections, we review the previous literature and develop our conceptual model by identifying the relationships among relevant constructs. We then describe the samples, data collection, and measures in the methodology section. Third, we report the results of the data analyses and the hypotheses tests. Finally, we discuss the findings and theoretical and managerial implications of this research. We also propose our limitations and directions for future research.

## 2. Literature Review and Hypotheses

### 2.1. Resource Complementarity, Relational Capital, and Alliance Performance

According to the resource-based theory, firms ally with partners for the sake of accessing complementary resources and stabilizing resource flows among different markets [4,20]. Resource complementarity captures the extent to which each partner brings unique strengths and resources of value to the collaboration [12,21]. Strategic alliances with a high level of resource complementarity are likely to promote deep integration and foster relational capital among partners since they would like to contribute more to achieve al-

liance goals. Relational capital refers to a relational rent produced in a relationship between organizations, and it is recognized as the sum of actual and potential resources embedded within the social network [22]. Three key dimensions of relational capital have been identified: communication, trust, and commitment [11,23]. Shared benefits from complementary resources motivate partnering firms to exchange information and reduce dysfunctional conflicts through open and frequent communications [12]. A high level of resource complementarity also indicates the reciprocal needs of partners which in turn decreases opportunism and enhances mutual trust in alliances [24]. In fact, resource complementarities between partners imply that the resources invested by both parties are inimitable and irreplaceable, making the partners more inclined to increase their interdependence [21], thereby enhancing value creation through relation-specific investments [25]. Moreover, alliance firms also demonstrate their expectations of a lasting relationship through the commitment to resources investments [12]. Resource-interdependent partners expect to maintain a solid relationship by committing specific asset investments to the relationship [11,25]. Based on these arguments, we propose hypothesis 1.

**Hypothesis 1 (H1).** *Resource complementarity is positively associated with relational capital.*

Relational capital can improve alliance performance through the joint contributions of the three dimensions including communication, trust, and commitment. Alliance performance is commonly conceptualized as the degree to which both partner firms achieve their strategic objectives in an alliance [26]. Communication is critical for improving alliance performance since it indicates the existing plans and future intentions of partners that lead the alliance to obtain success [27]. Frequent and open communication between partners is essential for understanding common goals [28], resolving inter-partner conflicts [23], and bolstering information exchanges [29]. Mutual trust can counteract overt self-interested behaviors, thereby curbing opportunism in the alliance [30]. Additionally, trust encourages both parties to jointly establish conventions and systems to reduce the ambiguity of knowledge transfer, which then facilitates the sharing of knowledge (especially tacit knowledge) [9,29]. Prior research has shown that information and knowledge exchanged in an alliance is probably inaccurate, incomplete, and out of date if there is a lack of mutual trust, because partners are hesitant to take the risk of sharing valuable and important information [31]. Commitment signals the willingness of warranting efforts to maintain an important ongoing relationship [32]. When firms obtain a commitment from their partners, they will likely invest and develop more relationship-specific assets in the alliance [33], leading to a higher alliance performance [34]. Additionally, commitment between partners can enhance integration and cooperation, which helps firms to achieve alliance goals. Empirical studies on supply chain alliances have confirmed the role of relationship commitment in the improvement of coordination and performance [35,36]. Taken together, sufficient relational capital with high levels of communication, trust, and commitment services is an effective way to achieve performance and long-term success in strategic alliances [9,37]. Hence, we propose hypothesis 2:

**Hypothesis 2 (H2).** *Relational capital is positively associated with alliance performance.*

As noted earlier, resource complementarity is associated with relational capital (H1), which in turn improves alliance performance (H2). We thus expect that relational capital is an important mechanism through which resource complementarity positively affects alliance performance. Prior literature suggests that accessing complementary resources encourages firms to form alliances, but the value of resource complementarity can be attenuated without appropriate relational governance [38]. In fact, interfirm collaboration can contribute to competitive advantages of alliances only when firms move beyond transaction-based trade and establish high-quality and long-term partnerships [22]. Thus, the reciprocal dependence resulting from resource complementarity motivates partners to build a solid cooperative relationship by investing relational capital to achieve their desired

outcomes [31,32]. Accordingly, alliance firms tend to increase inter-firm relational capital to improve the synergy of complementary resources, thereby leading to a superior alliance performance. Therefore, we propose hypothesis 3:

**Hypothesis 3 (H3).** *Relational capital mediates the relationship between resource complementarity and alliance performance.*

*2.2. The Moderating Effects of Environmental Dynamism and Environmental Hostility*

Dynamism and hostility are two important dimensions of environmental uncertainty and have received much attention in the relevant empirical literature due to their generalizability and quantifiability [38,39].

Environmental dynamism refers to the rate of change and unpredictability of change in an organization's environment [40]. In a dynamic environment, the timely and accurate acquisition and processing of environment-related information are key for firms to deal with uncertainty [41]. Investing large amounts of relational capital may render a firm relationally over-embedded and increase the risks of lock-ins which diminish the firm's adaptability to the volatile and changing environment [25]. Repeated ties can lead to information redundancies, constraining firms from exploring new market opportunities and novel knowledge [42,43]. An empirical study on the intertemporal alliance choices of SMEs has confirmed this finding by showing that firms in a highly dynamic environment tend to form short-term time-bound alliances in which flexible relational governance is adopted [44]. Research on vertical alliances also suggests that in a highly dynamic environment with rapid technological changes, providing a high level of commitment to partners is not optimal for alliance firms, and can even reduce the efficiency of cooperations [45,46]. The rationale is that increasing commitment to relation-specific investments in an alliance will constrain firms from exploring new opportunities [46]. Additionally, firms in alliances with high-resource complementarity are more likely to rely on the information and knowledge provided by their partners, while neglecting the necessary monitoring of the completeness and veracity of the information, thereby resulting in systematic bias [47,48]. Therefore, alliance firms in a dynamic environment are motivated to acquire non-overlapping resources and engage in knowledge exploration by investing in new partnerships, because a dynamic environment might erode the effectiveness and efficiency of internal resources in supporting firms to cope with external turbulence [49]. Taken together, it can be inferred that the positive effect of complementary resources on relational capital would be suppressed by environmental dynamism. We thus hypothesize:

**Hypothesis 4a (H4a).** *The relationship between resource complementarity and relational capital is negatively moderated by environmental dynamism: the positive resource complementarity–relational capital link is weaker when environmental dynamism is high than when it is low.*

On the other hand, environmental hostility captures the level of threat from competition and the availability of opportunities and resources in the external environment [50,51]. Firms operating in a hostile environment are often inclined to avoid excessive risk-taking, because taking extra risks is especially hazardous in highly competitive conditions [52]. As environmental hostility rises, resources available for developing and introducing new products declines [53]. In order to achieve a competitive advantage in a hostile environment, firms are willing to invest in the existing valuable partnerships to constantly obtain complementary resources that are rare in the market [54]. Additionally, strengthening the existing cooperative relationships by investing relational capital allows firms to receive support at relatively low costs, and set up entry barriers in reacting to environmental hostility [11]. For example, previous research has suggested that partners with a stable and trusting relationship can fully exploit the resources available to them, thus reducing the costs of inventory, production, and transportation [55]. Second, a partnership based on trust, commitment, and communication can improve the efficiency of cooperation among

alliance firms, especially in a hostile environment. Highlighting the harmonious relationships with their complementary partners can also improve consumers' trust in the firm, and thus lead to more repeated purchase behaviors and positive WOMs [56]. A study on international marketing channels indicates that exporting manufacturers are motivated to use relational norms and increase relational capital to achieve competitiveness when threatened by the hostility of the export market [57]. This finding indicates that relational capital can convey positive signals to the market, which in turn enhances firms' cooperative reputation or poses a threat to their revivals. Taken together, environmental hostility reinforces the positive influence of resource complementarity on relational capital. Therefore, we propose hypothesis 4b.

**Hypothesis 4b (H4b).** *The relationship between resource complementarity and relational capital is positively moderated by environmental hostility: the positive resource complementarity–relational capital link is greater when environmental hostility is high than when it is low.*

### 2.3. The Moderated Mediation Model

The above arguments suggest an integrated theoretical framework in which relational capital mediates the positive relationship between resource complementarity and alliance performance and environment uncertainty moderates the positive connection between resource complementarity and relational capital. We further theorize a moderated mediation model linking the interactive effects of resource complementarity and (a) environmental dynamism, and (b) environmental hostility through relational capital. As previously mentioned, alliance firms with the complementary resource are more likely to invest relational capital that benefit alliance performance in a less dynamic or more hostile environment. Hence, we expect a stronger positive indirect relationship between resource complementarity and alliance performance when the environment is featured with low dynamism or high hostility, because firms tend to maintain a more stable relationship with their current partners in these conditions, and thus achieve higher alliance performance. On the contrary, when the environment is more dynamic or less hostile, alliance firms invest a lower level of relational capital in existing alliances to stay flexible, thus weakening the positive indirect link between resource complementarity and alliance performance. In sum, we propose the following hypotheses:

**Hypothesis 5a (H5a).** *The indirect relationship between resource complementarity and alliance performance through relational capital is moderated by environmental diversity, such that this indirect effect is stronger when the environment is less dynamic.*

**Hypothesis 5b (H5b).** *The indirect relationship between resource complementarity and alliance performance through relational capital is moderated by environmental hostility, such that this indirect effect is stronger when the environment is more hostile.*

The conceptual model of this research is illustrated in Figure 1.

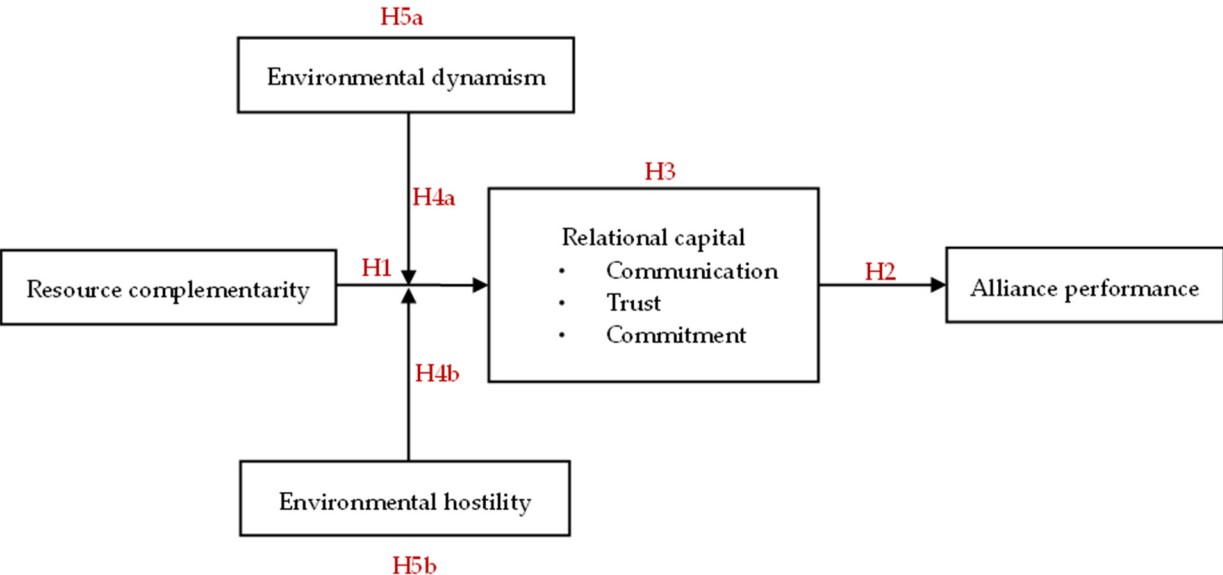

**Figure 1.** Conceptual model.

### 3. Methodology

*3.1. Research Design, Samples and Data Collection*

We adopted a survey of alliance firms operating in China to test the above hypotheses. This study chose China as the research context for several reasons: First, firms in China increasingly fulfill important strategic objectives by establishing strategic alliances [58]. Second, business activities in China are intensively influenced by social relations, and this phenomenon provides an appropriate social context to study strategic alliances from a relational perspective [8]. Third, transitional economies like that of China are featured with high-uncertainty characteristics, which allowed us to investigate the role of environmental uncertainty in determining the outcomes of strategic alliances [59].

A pilot study was conducted on a group of 55 alliance firms. We collected their feedback on the readability and applicability of the questionnaire, and further modified and refined some items. The formal questionnaires were distributed to 338 EMBA/MBA students enrolled in executive business programs at several eastern universities in China and 51 senior managers who were past project partners of our research team. Previous literature pointed out that the EMBA or MBA student samples could be appropriate for business research when they were involved in high-level decision-making processes [60]. The data collection took place over four months and received an effective response rate of 84.6% for the EMBA/MBA students (collected in the first round) and 84.3% for the senior managers (collected in the second round). We identified samples relying on two criteria: (1) the firm in which the informant worked had alliance experience, and (2) the informant was a senior leader who had extensive experience of managing alliance projects in the firm and was involved in high-level decision-making processes in the alliance projects. The respondents who did not meet these criteria were removed from the sample. We eliminated invalid or incomplete responses from the analyses, resulting in a total of 210 valid questionnaires. Participants were asked to fill out the questionnaire according to the situation of the recent alliance they were familiar with. We found no significant difference between the responses received in the first round and that in the second round on the demographic characteristics or other key variables.

The majority of the firms were located in the Yangtze River Delta region of China (88.6%), others were in the Pearl River Delta region (4.8%) and Bohai Sea Economic Zone (4.8%), and so on. Foreign-owned firms contributed the largest proportion (44.3%), followed by state-owned (22.9%) and private-owned (20.5%). Nearly half of the firms had over 1000 employees (46.2%). The firms were diverse in terms of industry: manufacturing (44.3%),

information technology and services (14.8%), financial and insurance (8.6%), transportation (7.6%), retailing (6.7%), as well as electrical and electronics (5.7%). The alliance types included equity alliance (34.3%), co-marketing (26.2%), R&D alliance (16.7%), co-production (12.4%), and OEM (10.4%).

### *3.2. Measures*

All variables were measured with established scales in prior literature, in which some items were refined based on the feedback collected in the pilot study. These items were ranked along 7-point scales (1= strongly disagree, 7 = strongly agree). The questionnaire was developed in English and two bilingual researchers translated it into Chinese using the back-translation procedure [61].

### 3.2.1. Dependent Variable

*Alliance performance.* Given the majority of alliances seldom report objective financial performance indices, many scholars used subjective measures to assess firm-level alliance performance, such as an evaluation of the fulfillment of the firm's strategic alliance goals [47]. In the present research, alliance performance was measured with a five-item scale adapted from previous alliance research [62,63]. The scale captured the firms' satisfaction with the alliance outcomes ("overall, we are very satisfied with the performance of this alliance"), the degree of goal achievement ("the alliance has realized the goals we set out to achieve"), and its value contributed to the firms ("the strategic alliance improves profitability," "the strategic alliance accesses skills and learning for future competitive advantages", and "the strategic alliance helps us to realize our business objectives"). We averaged the five items as an overall alliance performance indicator.

### 3.2.2. Independent Variable

*Resource complementarity.* We employed a four-item scale to measure resource complementarity, including the following items: "we have complementary strengths that are useful to our relationship" "our partner brings to the table resources and competencies that complement our own", "we both contribute complementary resources to the relationship that help us achieve mutual goals", "our partner provides resources and capacities that we need" [64–66].

*Environmental dynamism.* Consistent with the existing literature [38,58], environmental dynamism was measured on a three-item scale: "market demand and consumer tastes have been unpredictable", "our clients regularly ask for new products and services", and "environmental changes in our local market are intense".

*Environmental hostility.* We measured environmental hostility with an adapted version of the scale developed by [38,67]. The scale included four items describing the extent of competition in the market: "competition in our market is intense", "our firm has relatively strong competitors", "competition in our market is extremely high", and "market activities of our competitor is unpredictable".

*Relational capital.* Relational capital consists of three dimensions, including communication, trust, and commitment. We measured the quality and frequency of communication from the following aspects: "timely", "complete", and "frequent" [47,68]. Trust was measured using six items such as "keep the promise", "treat fairly", "consider the effects of its decisions and actions on the partner", "consider each other's interests" "integrity" and "honest" [46,69,70]. Commitment was measured with four items, such as "intend to maintain the relationship", " deserve efforts to maintain the relationship", "willing to make the long-term investment", and "very committed to the relationship" [32,46]. Consistent with [11], we adopted an internal-consistency approach to parcel the items under each dimension of communication, trust, and commitment. These three parcels were generated as observed indicators to form the latent variable of relational capital.

### 3.2.3. Control Variable

This study included several control variables such as firm size, firm ownership, firm age, and alliance type in the model. Firm size was measured by the number of employees. Firm ownership included state-owned, private-owned, foreign-owned, and sino-foreign joint ventures. Firm age was reflected by the number of years the firm had been in operation. Alliance types ranged from joint venture, cross-ownership, R&D alliance, OEM, co-production, co-marketing, and franchising.

## 4. Results

### 4.1. Reliability and Validity

We assessed the scale reliability by computing Cronbach's alpha coefficients. Results showed that Cronbach's alpha coefficients for all scales were above the recommended minimum of 0.6, indicating an acceptable reliability [71]. Convergent validity was adequate with all AVE (average variance extracted) values reaching 0.4 and composite reliability (C.R.) values reaching 0.7 [72]. The AVE's square root for any pair of constructs exceeded their correlation, which confirmed the discriminant validity of our measures. Table 1 displays the assessments of reliability and validity. Table 2 displays the descriptive statistics and correlations of the variables. In addition, several fit indices supported a good fitness of the model: $\chi^2/\mathrm{d}f$ = 1.700, GFI = 0.895, NFI = 0.842, IFI = 0.928, CFI = 0.927, RMR = 0.046 and RMSEA = 0.058.

**Table 1.** Assessment of reliability and validity.

| Variables | Items | Cronbach's $\alpha$ | C.R. | Factor Loading | AVE |
|---|---|---|---|---|---|
| Resource Complementarity | 4 | 0.709 | 0.729 | 0.607–0.768 | 0.411 |
| Environmental Dynamism | 3 | 0.761 | 0.760 | 0.759–0.804 | 0.515 |
| Environmental Hostility | 4 | 0.783 | 0.857 | 0.655–0.811 | 0.475 |
| Communication | 3 | 0.688 | 0.754 | 0.570–0.816 | 0.535 |
| Trust | 6 | 0.907 | 0.884 | 0.675–0.831 | 0.607 |
| Commitment | 4 | 0.824 | 0.820 | 0.626–0.828 | 0.545 |
| Alliance Performance | 5 | 0.884 | 0.914 | 0.703–0.863 | 0.607 |

Note: C.R. = composite reliability, AVE = average variance extracted.

**Table 2.** Correlations, means, and standard deviations.

| | RSC | EVD | EVH | CMU | TRS | CMI | APF |
|---|---|---|---|---|---|---|---|
| RSC | 0.641 | | | | | | |
| EVD | 0.073 | 0.718 | | | | | |
| EVH | 0.264 ** | 0.388 ** | 0.689 | | | | |
| CMU | 0.406 ** | −0.041 | 0.132 | 0.731 | | | |
| TRS | 0.147 * | −0.185 ** | 0.149 * | 0.258 ** | 0.779 | | |
| CMI | 0.254 ** | 0.231 ** | 0.231 ** | 0.323 ** | 0.289 ** | 0.738 | |
| APF | 0.279 ** | −0.134 | 0.129 | 0.307 | 0.517 ** | 0.334 ** | 0.779 |
| Mean | 3.920 | 3.213 | 3.900 | 3.706 | 3.454 | 3.791 | 3.655 |
| S.D. | 0.649 | 0.828 | 0.664 | 0.696 | 0.816 | 0.648 | 0.799 |

Note. The diagonal values are the square roots of the AVE for each construct. N = 210, ** $p < 0.01$, * $p < 0.05$ (two-tailed *t*-test). RSC = resource complementarity; EVD = environmental dynamism; EVH = environmental hostility; RLC = relational capital; PWD = power dependence; APF = alliance performance.

### 4.2. Assessment of Common Method Bias (CMV)

We adopted some procedural remedies to alleviate common method variance concerns [73]. We guaranteed that participants' answers would be anonymous and confidential, and stated that there were no right or wrong answers. To improve the readability and comprehensibility of the questionnaire, we constructed survey questions to avoid vague terms and complicated syntax. We also counterbalanced the order of all items of variables

in the questionnaire. Furthermore, we employed multiple post hoc statistical methods to assess the threat of CMV. First, we adopted the Harman's single-factor test [73,74], producing eight factors, the largest of which extracted only 23.069% of the variance. Second, we followed Podsakoff et al.'s (2012) recommendations, comparing the fit indices of the single-factor CFA model with the original measurement CFA model [75]. Results showed that the single-factor CFA model exhibited a poorer fit than the measurement model (Model$_{single-factor}$: $\chi^2/\mathrm{d}f$ = 5.207, GFI = 0.537, NFI = 0.379, IFI = 0.430, CFI = 0.424, RMR = 0.114, RMSEA = 0.142; $\Delta\chi^2$ = 1356.466, $\Delta\mathrm{d}f$ = 22, $p < 0.01$). Third, we used the unmeasured latent method construct (ULMC) technique ([75,76] to control for the potential effect of CMV. The model with the method factor was compared to the identical model except with construct correlations constrained to the values obtained in the original measurement model [77]. This analysis suggested that the fit of the two models revealed no substantive differences ($\Delta\chi^2$ = 19.920, $\Delta\mathrm{d}f$ = 21, *ns.*). Taken together, it can be concluded that common method bias was not a major issue in this study.

*4.3. Hypotheses Tests*

We performed a series of hierarchical multiple regression analyses to test the hypotheses. As recommended by [78]. we mean centered all of the independent variables prior to the formation of interaction terms. No significant multicollinearity problems were found with the assessments of variance inflation factors (VIFs range from 1 to 2). Table 3 presents the results of the hierarchical regression analysis. Results show that resource complementarity had a positive relationship with relational capital (Model 1b: $\beta$ = 0.366, $p < 0.01$). Therefore, Hypothesis 1 was supported. Next, we examined the relationship between relational capital and alliance performance. Model 2b indicates that relational capital was significantly positively related to alliance performance (Model 2b: $\beta$ = 0.564, $p < 0.01$), providing support for Hypothesis 2. Furthermore, we tested the mediating role of relational capital between the relationship of resource complementarity and alliance performance, following the procedure suggested by [79]. In the first step, we regressed the dependent variable of alliance performance on the independent variable of resource complementarity, resulting in a positive relationship between resource complementarity and alliance performance (Model 2c: $\beta$ = 0.274, $p < 0.01$); Second, it has been testified that the mediator, named relational capital, had a positive relationship with the independent variable of resource complementarity as well as the dependent variable of alliance performance. The third step included relational capital in Model 2d as the mediator, and found that the effect of resource complementarity on alliance performance was eliminated (Model 2d: $\beta$ = 0.080, $p > 0.05$), while the effect of relational capital was significant ($\beta$ = 0.532, $p < 0.01$). Therefore, the mediating role of relational capital has been validated preliminary. In addition, we also used the bootstrapping method to estimate the indirect effects of the independent variable [80]. This analysis was conducted based on a sample size of 5000. The results revealed that the direct effect of resource complementarity by controlling the mediator does not reach significance, with a 95% confidence interval (CI) of −0.056 to 0.252 which includes zero. However, the indirect effect through relational capital was significant and the bootstrapping 95% confidence interval (CI) excludes zero [0.160, 1.335], indicating that relational capital mediates the relationship between resource complementarity and alliance performance (see Table 4). Therefore, Hypothesis 3 was supported.

In Model 1c, we included the two-way interaction term between resource complementarity and environmental dynamism as well as the interaction between resource complementarity and environmental hostility to Model 1b (Model 1b:$R^2$= 0.415, F = 8.478, $p < 0.01$; Model 1c:$R^2$= 0.506, F = 7.631 **, $p < 0.01$). The interactive effect of resource complementarity and environmental dynamism was significantly negative (Model 1c: $\beta$ = − 0.207, $p < 0.01$). To better understand the moderate effect of environmental dynamism, we plotted the interaction at one standard deviation (SD) above and one standard deviation (SD) below the mean of environmental dynamism. The result showed that resource complementarity was less positively associated with relational capital in the situation of higher environmen-

tal dynamism than that of lower environmental dynamism (see Figure 2), which supported Hypothesis 4a. We further employed floodlight analysis using the Johnson–Neyman technique to identify the ranges of significance, where the turning point of environmental dynamism was at a value of 3.78 ($b_{JN}$ = 0.21, $SE$ = 0.10, $p$ = 0.05). The results again confirmed Hypothesis 4a by suggesting that relational capital did not change with the increase in resource complementarity between partners at high levels of environmental dynamism (above 3.78), while resource complementarity positively influenced relational capital when the environment was less dynamic (below 3.78). The finding is presented in Figure 3. However, the interactive effect of resource complementarity and environmental hostility did not reach significance (Model 1c: $\beta$ = 0.063, $p$ > 0.05), thus Hypothesis 4b was not supported.

**Table 3.** Results of hierarchical multiple regression analyses.

| Variables | Relational Capital | | | Alliance Performance | | | |
|---|---|---|---|---|---|---|---|
| | Model 1a | Model 1b | Model 1c | Model 2a | Model 2b | Model 2c | Model 2d |
| Control variables | | | | | | | |
| Firm age | −0.101 | −0.060 | −0.056 | −0.044 | 0.009 | −0.013 | 0.019 |
| Firm size | 0.162 | 0.109 | 0.095 | 0.080 | −0.014 | 0.040 | −0.018 |
| Firm ownership | 0.068 | 0.095 | 0.080 | −0.060 | 0.071 | −0.040 | −0.091 |
| Alliance types | 0.162 | 0.109 | 0.095 | 0.080 | 0.026 | 0.040 | −0.018 |
| Main effects | | | | | | | |
| Resource complementarity | | 0.366 ** | 0.298 ** | | | 0.274 ** | 0.080 |
| Environmental dynamism | | | −0.183 ** | | | | |
| Environmental hostility | | | 0.231 ** | | | | |
| Mediating effect | | | | | | | |
| Relational capital | | | | | 0.564 ** | | 0.532 ** |
| Moderating effects | | | | | | | |
| Resource complementarity × Environmental dynamism | | | −0.207 ** | | | | |
| Resource complementarity × Environmental hostility | | | 0.063 | | | | |
| $R^2$ | 0.205 | 0.415 | 0.506 | 0.008 | 0.313 | 0.284 | 0.315 |
| Adjusted $R^2$ | 0.023 | 0.152 | 0.222 | −0.012 | 0.296 | 0.058 | 0.294 |
| $F$ | 2.248 | 8.478 ** | 7.631 ** | 0.398 | 18.588 ** | 3.582 ** | 15.539 ** |

Note: Standardized coefficients are reported. ** $p$ < 0.01 (two-tailed $t$-test).

　　Hypotheses 5a and 5b propose that the mediating effect of relational capital differs depending on environmental dynamism and environmental hostility. We performed a bootstrapping procedure to quantify the indirect effect of resource complementarity on alliance performance via relational capital at low, mean, and high levels of the two moderators. As shown in Table 4, the indirect effect of resource complementarity was significant and none of the bootstrapping 95% confidence intervals contained zero at a low (−1SD) rather than a high level (+1SD) of environmental dynamism, regardless of the values of environmental hostility. In line with H5a, the mediating effect of relational capital was moderated by environmental dynamism, and it was enhanced when the environment was less dynamic. On the other hand, the indirect effect of resource complementarity via relational capital did not vary with the values of environmental hostility, not supporting H5b. One possible explanation is that fierce competition may make firms near-sighted and opportunistic. As a result, alliance firms in a hostile environment may be driven by short-term benefits and engage in opportunistic behaviors such as improper exploitation of partners' complementary resources, which in turn attenuates the positive effect of resource compatibility.

**Table 4.** Bootstrapping results of mediation and moderated mediation tests.

| Path | | Effect | BootSE | LLCI | ULCI |
|---|---|---|---|---|---|
| Mediation Model: Direct effect RSC→APF | | 0.098 | 0.078 | −0.056 | 0.252 |
| indirect effect RSC→RLC→APF | | 0.439 | 0.363 | 0.160 | 1.335 |
| **Moderated Mediation Model** | | | | | |
| EVD | EVH | Effect | BootSE | LLCI | ULCI |
| −0.828(-1SD) | −0.665(-1SD) | 0.299 | 0.071 | 0.168 | 0.447 |
| −0.828(-1SD) | 0.000(Mean) | 0.337 | 0.076 | 0.203 | 0.499 |
| −0.828(-1SD) | 0.665(+1SD) | 0.375 | 0.098 | 0.206 | 0.585 |
| 0.000(Mean) | −0.665(-1SD) | 0.159 | 0.069 | 0.029 | 0.295 |
| 0.000(Mean) | 0.000(Mean) | 0.197 | 0.059 | 0.098 | 0.326 |
| 0.000(Mean) | 0.665(+1SD) | 0.235 | 0.073 | 0.112 | 0.398 |
| 0.828(+1SD) | −0.665(-1SD) | 0.019 | 0.110 | −0.205 | 0.231 |
| 0.828(+1SD) | 0.000(Mean) | 0.057 | 0.095 | −0.124 | 0.249 |
| 0.828(+1SD) | 0.665(+1SD) | 0.095 | 0.094 | -0.077 | 0.294 |

Note: N = 210. Bootstrap sample size = 5000. SE = standard error. LL = lower limit; UL = upper limit; CI = confidence interval. 95% confidence interval.

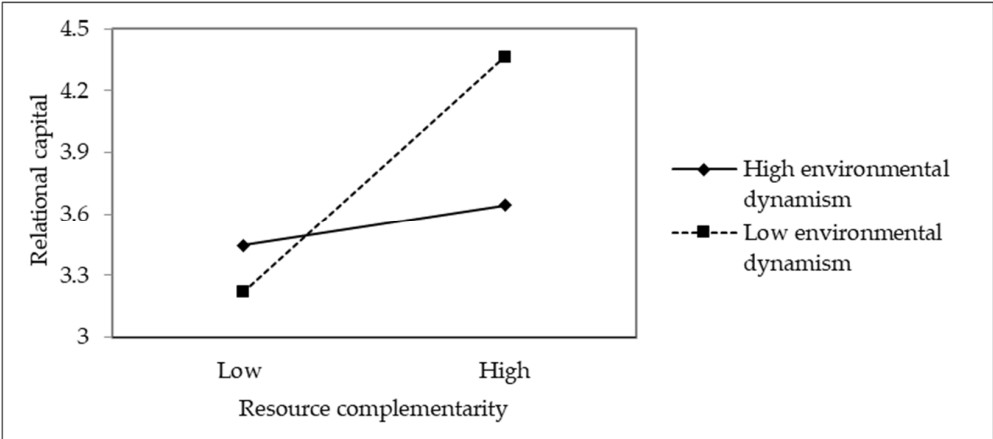

**Figure 2.** The moderating effect of environmental dynamism on the relationship between resource complementarity and relational capital.

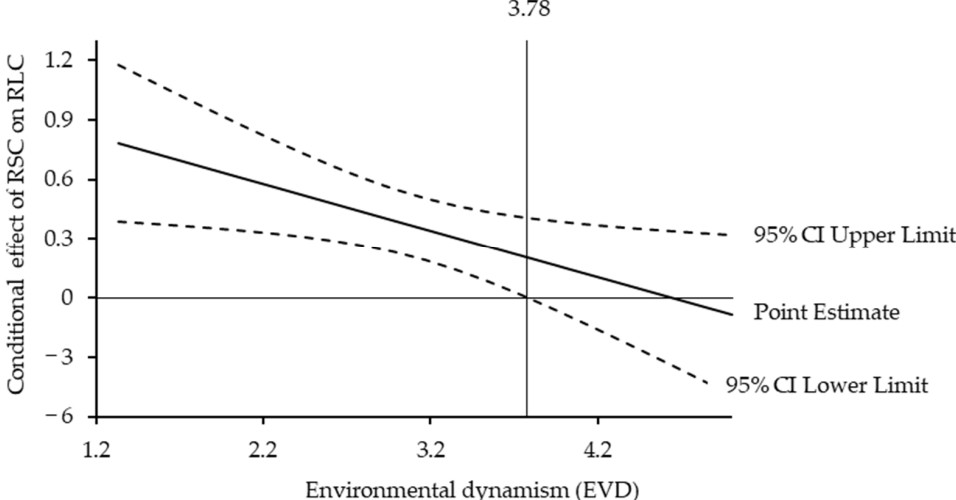

**Figure 3.** Floodlight analysis results for the resource complementarity × environmental dynamism interaction effect on alliance performance.

## 5. Discussion

This study developed an integrative theoretical framework in which we examined how and when resource complementarity affects strategic alliance performance via relational capital in an uncertain environment with a moderated mediation approach. The empirical results indicate that complementary resources a firm can access from its strategic alliance motivate it to invest relational capital in the partnership, which in turn improves alliance performance. However, the effect of resource complementarity on relational capital is diminished under a highly dynamic environment because firms need to keep flexibility and lower lock-in risks in a volatile market. Furthermore, the moderated mediation model accounts for how and under which conditions firms can better leverage relational capital to achieve alliance success. Based on the moderated mediation analysis, we found that the mediating effect of relational capital between resource complementarity and alliance performance was stronger when the environment was less dynamic. Results of this study have significant theoretical and managerial implications.

### 5.1. Theoretical Contributions

The present study explored the relationships among multiple theories (e.g., resource-based theory, the relational view, and the contingency theory) to provide a comprehensive explanation of alliance outcomes from both inter-organizational and external environmental perspectives. Specifically, our research highlights that both resources and relationships are key drivers of alliance success, providing empirical evidence for the interpretations of the resource-based theory and the relational view for predicting alliance outcomes [81,82]. Second, we bridged these two theories by showing how resource complementarity improves alliance performance through the underlying mechanism of relational capital, thereby answering the calls for combining the structural approach with the sociopsychological approach in analyzing alliance outcomes [12,13]. Moreover, although some scholars have attempted to incorporate inter-organizational and external environmental factors in predicting alliance success [11,83], their work does not demonstrate how these factors interact to affect the relationship and the outcomes of collaboration. We extended their research by exploring the interaction of resource complementarity and environmental uncertainty on relational capital and alliance performance with a moderated mediation analysis. Findings regarding the impact of resource complementarity on relational capital as well as the boundary conditions highlight the applicability of the resource-based theory and expand our understanding of the effectiveness of complementary resources in different environmental conditions. In addition, this research contributes to the contingency theory and the environmental uncertainty literature by examining whether and how two types of environmental uncertainty can shape partnerships and alliance performance. According to the contingency theory, it is essential for firms to take contextual factors (e.g., environmental uncertainty) into account when designing alliance structure, and firms should achieve proper strategic fits between alliance forms and the external environment to create synergistic benefits [84,85]. However, extant studies have suggested inclusive conclusions regarding the effect of environmental uncertainty on the formation and the outcome of partnerships in alliances [11,48,86–88]. While some literature indicates that alliance firms should reduce their dependency on partnerships in an uncertain environment to maintain flexibility in switching partners to explore more opportunities [48], other research asserts that firms in an uncertain environment have higher alliance motivations that drive them to increase relational capital to access complementary resources and reduce transaction costs [11,88]. We argue that these conflicting conclusions may be due to disparate effects of different types of environmental uncertainty. Our results indicate that the indirect effect of complementary resources on alliance performance through relational capital is contingent on environmental uncertainty, such that this effect is enhanced in a less dynamic rather than a less hostile environment. Therefore, this research offers a potential explanation in an attempt to reconcile the different findings within the previous studies on the effect of environmental uncertainty in the domain of strategic alliance.

### 5.2. Managerial Implications

Several important implications emerge for managing strategic alliances in an uncertain environment. First, this study indicates that allying with firms with complementary resources can increase relation capital and thereby achieve a higher alliance performance, which offers alliance managers an important criterion for choosing appropriate partners. Moreover, the mediating role of relational capital highlights the fact it is difficult for alliance firms to benefit from complementary resources without adequate relational capital. Thus, alliance firms should take advantage of relational-based governance relying on communication, trust, and commitment as a complement to the contract-based mechanism. Third, we shed light on the complicated influences of environmental uncertainty. Accordingly, alliance firms should consciously and continuously scan the external environment to detect different types of uncertainty. It is especially important to introduce a contingency perspective to the management of strategic alliances so that firms can respond to environmental uncertainty in a timely and effective manner.

### 5.3. Limitations and Directions for Future Research

While this study makes important contributions to the alliance literature, several limitations should be pointed out. First, our findings are based on cross-sectional survey data, which limits our understanding of the dynamic nature of the variables. It would be beneficial to conduct a longitudinal study to further validate the hypothesized relationships with the dynamic evolutions of strategic alliances. Second, although we collected data via different channels and ensured that the sampled firms came from various industries and areas, the generalization of the results may be constrained using a convenience sampling method. Future research involving larger random samples would help to replicate our results. Like most survey research on alliances, this study collected responses from one side of the alliance owing to special partnership agreements [23,89]. We could have provided stronger evidence if a dyadic data collection approach is adopted. Moreover, although our study showed the significant impacts of internal resources and their interactions with the external environment on relational capital and alliance performance, we did not examine the effect of organizational culture, which is also a very important factor in challenging inter-firm cooperation. Accordingly, taking organizational culture into account in our framework can provide a deeper understanding of the full picture of relational governance and alliance outcomes.

**Author Contributions:** Conceptualization, X.L. and Y.S.; methodology, X.L.; software, X.L. and W.W.; validation, Y.S.; formal analysis, X.L. and W.W.; investigation, X.L. and Y.S.; writing—original draft preparation, X.L.; writing—review and editing, Y.S. and W.W.; supervision, X.L. and Y.S.; project administration, X.L.; funding acquisition, X.L. All authors have read and agreed to the published version of the manuscript.

**Funding:** This research was funded by the National Natural Science Foundation of China, grant number: 71902060; Natural Science Foundation of Hunan Province, China, grant number: 2020JJ5375; Research Foundation of the Education Department of Hunan Province, China, grant number: 19C1156.

**Data Availability Statement:** Data is contained within the article.

**Conflicts of Interest:** The authors declare no conflict of interest.

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
