# Peer review of "Leveraging Complementary Resources through Relational Capital to Improve Alliance Performance under an Uncertain Environment: A Moderated Mediation Analysis"

_sustainability, doi:10.3390/su15010310_

Round 1
Reviewer 1 Report
An interesting paper which should be improved before publishing.
1. The study results do not show how and when complementary complementary resource affects alliance performance through the mechanism of relational capital in an uncertain environment. Therefore, the specific objective should be changed. For example: "we analyse the effect of complementary resources on alliance performance through….”(abstract).
2. The introduction was not developed to position the article in relation to previous research. This should be done.
3. The paper demonstrates an adequate understanding of the relevant literature in the research topic and cites appropriate literature sources. However, the authors should present definitions of the main research topics: strategic alliances; resource complementarity; alliance performance; environmental dynamism; environmental hostility. In addition, the justification of research hypotheses should be improved (based on theories and previous studies).
4. The authors should put the research hypotheses in the conceptual model (figure 1)
5. The authors indicate that “formal questionnaires were distributed to 338 EMBA/MBA students enrolled in the executive business programs at several eastern universities in China and 51 senior managers who were the past project partners of our research team. Previous literature pointed out that the EMBA or MBA student samples could be appropriate for business research when they were involved in the high-level decision-making process”. I ask whether all EMBA or MBA students were involved in (high-level) decision-making processes in alliance companies? This should be clear from the paper.
6. In section 5 - Discussion, the authors should discuss the results, which implies comparing the results of the study with previous studies.
7. The conclusions do not show the practical contributions of the study.
8. Review title of paper to suit content.
Reviewer 2 Report
The article Leverage complementary resources through relational capital in strategic alliances under an uncertain environment: a moderated mediation analysis develops an integrative theoretical approach that examines how and when complementary resources affect the effectiveness of strategic alliance through relational capital in an uncertain environment with a moderated mediation approach.
Empirical results based on data from a survey of 210 alliance firms show that the additional resources a firm can access through its strategic alliance encourages it to invest relational capital in the partnership, which in turn increases the effectiveness of the alliance.
Based on moderated mediation analysis, it is found that the mediating effect of relational capital between resource complementarity and alliance efficiency is stronger when the environment is less dynamic.
The results of this study have important theoretical and managerial implications.
Although this study contributes significantly to the literature on alliances, the main observation, to the results obtained in the paper, is that the study, uses empirical data from a survey of 210 alliance firms, through different channels and the authors made sure that the firms included in the sample were from different industries and areas, and the generalization of results was limited to using a convenient sampling method.
Nevertheless, it is necessary to conduct the study with a larger random sample in order to replicate the important results obtained.
Reviewer 3 Report
The introduction part doesn't clearly state what is the original contribution of this paper is.
The introduction part should in the end explain the research flow of the paper.
The literature review is a bit short considering the number of hypotheses in this paper.
You can look at the structure of this paper which could help in improving of your paper:
Effect of open-mindedness and humble behavior on innovation: mediator role of learning. International Journal of Emerging Markets
Round 2
Reviewer 1 Report
I am satisfied with the response of the authors